# The *TNFRSF13C* H159Y Variant Is Associated with Severe COVID-19: A Retrospective Study of 500 Patients from Southern Italy

**DOI:** 10.3390/genes12060881

**Published:** 2021-06-08

**Authors:** Roberta Russo, Immacolata Andolfo, Vito Alessandro Lasorsa, Sueva Cantalupo, Roberta Marra, Giulia Frisso, Pasquale Abete, Gian Marco Cassese, Giuseppe Servillo, Gabriella Esposito, Ivan Gentile, Carmelo Piscopo, Matteo Della Monica, Giuseppe Fiorentino, Giuseppe Russo, Pellegrino Cerino, Carlo Buonerba, Biancamaria Pierri, Massimo Zollo, Achille Iolascon, Mario Capasso

**Affiliations:** 1Dipartimento di Medicina Molecolare e Biotecnologie Mediche, Università degli Studi di Napoli Federico II, 80131 Napoli, Italy; roberta.russo@unina.it (R.R.); andolfo@ceinge.unina.it (I.A.); lasorsa.alessandro@gmail.com (V.A.L.); sueva.cantalupo@tiscali.it (S.C.); robertamarra.r@gmail.com (R.M.); gfrisso@unina.it (G.F.); gabriella.esposito@unina.it (G.E.); massimo.zollo@unina.it (M.Z.); achille.iolascon@unina.it (A.I.); 2CEINGE Biotecnologie Avanzate, 80145 Napoli, Italy; 3COVID Hospital, P.O.S. Anna e SS. Madonna della Neve di Boscotrecase, Ospedali Riuniti Area Vesuviana, 80042 Boscotrecase, Italy; abete2002@libero.it (P.A.); cassesegianmarco@gmail.com (G.M.C.); 4Dipartimento di Neuroscienze e Scienze riproduttive ed odontostomatologiche, Università degli Studi di Napoli Federico II, 80131 Napoli, Italy; giuseppe.servillo@unina.it; 5Dipartimento di Medicina Clinica e Chirurgia, Università degli Studi di Napoli Federico II, 80131 Napoli, Italy; ivan.gentile@unina.it; 6Medical and Laboratory Genetics Unit, A.O.R.N. ‘Antonio Cardarelli’, 80131 Napoli, Italy; carmelo.piscopo@aocardarelli.it (C.P.); matteo.dellamonica@aocardarelli.it (M.D.M.); 7AORN dei Colli Presidio Ospedaliero Cotugno, 80131 Napoli, Italy; giuseppefiorentino1@gmail.com; 8Unità di Radiologia, Casa di Cura Villa dei Fiori, 80011 Acerra, Italy; dott-russo@libero.it; 9Istituto Zooprofilattico Sperimentale del Mezzogiorno, 80055 Portici, Italy; strategia@izsmportici.it (P.C.); carlo.buonerba@izsmporticit.it (C.B.); biancamaria.pierri@izsmportici.it (B.P.); 10Dipartimento di Medicina, Chirurgia e Odontoiatria “Scuola Medica Salernitana”, Università di Salerno, 84081 Baronissi, Italy

**Keywords:** COVID-19, SARS-CoV-2, whole-exome sequencing, SNP genotyping, CVID, *TNFRSF13C*

## Abstract

To identify host genetic determinants involved in humoral immunity and associated with the risk of developing severe COVID-19, we analyzed 500 SARS-CoV-2 positive subjects from Southern Italy. We examined the coding sequences of 10 common variable immunodeficiency-associated genes obtained by the whole-exome sequencing of 121 hospitalized patients. These 10 genes showed significant enrichment in predicted pathogenic point mutations in severe patients compared with the non-severe ones. Moreover, in the *TNFRSF13C* gene, the minor allele of the p.His159Tyr variant, which is known to increase NF-kB activation and B-cell production, was significantly more frequent in the 38 severe cases compared to both the 83 non-severe patients and the 375 asymptomatic subjects further genotyped. This finding identified a potential genetic risk factor of severe COVID-19 that not only may serve to unravel the mechanisms underlying the disease severity but, also, may contribute to build the rationale for individualized management based on B-cell therapy.

## 1. Introduction

Up to May 2021, the coronavirus disease 2019 (COVID-19) pandemic due to SARS-CoV-2 infection caused approximately 160 million confirmed cases and more than three million deaths (https://covid19.who.int/, accessed on 17 May 2021). About 15% of cases develop severe pneumonia with acute respiratory distress syndrome, which often necessitates mechanical ventilation (MV), dysregulated immune response, systemic inflammation, and cytokine storm [1,2].

Advanced age, male gender, and chronic comorbidities can worsen the clinical disease course. However, the host genetic background also plays a role in disease variability. From this perspective, the determinants of different susceptibilities to SARS-CoV-2 mostly involve genes related to the initial stages of infection, as demonstrated for genetic variants of the *ACE2* and *TMPRSS2* genes [3,4,5,6]. Furthermore, common variants at diverse loci (3p21.31, 9q34.2, 19p13.3, 12q24.13, and 21q22.1) and inactivating rare mutations in genes belonging to the type I interferon pathway have been found to associate with severe COVID-19 [7,8,9]. 

Conversely, the determinants of variable COVID-19 severity mainly include components of the host immune response to the infection. Approximately 40 genes have been associated with a susceptibility to SARS-CoV-2 infection and 50% of them with disease severity. They are involved in multiple immunological pathways including either the innate or the adaptive immune responses [2,10]. Regarding an innate immunity, several studies highlighted the importance of the intact type I interferon pathway, as well as of immune signaling pathways, including Toll-like receptors [10]. Instead, for adaptive immunity, the impact of primary immunodeficiencies associated with antibody deficiencies on a COVID-19 outcome is not yet clarified. A recent survey on primary antibody deficiencies described a severe clinical course in patients with common variable immunodeficiency (CVID) compared to patients with pure agammaglobulinemia. This observation suggests a contribution of immune dysregulation, including dysfunctional B cells, to the severe inflammatory phenotype in CVID [11].

We herein investigated the host genetic factors associated with COVID-19 severity by analyzing a case series of 500 patients from Campania, a region of Southern Italy, where the pandemic caused 325,178 infections and 6816 deaths by May 2021. A targeted analysis of whole-exome sequencing (WES) data highlighted a mutational enrichment of CVID-associated genes in severe patients compared to non-severe ones. In particular, severely affected subjects showed a recurrent rare variant, p.His159Tyr (H159Y), in the *TNFRSF13C* gene, encoding the B cell-activating factor receptor (BAFFR) [12]. These findings contribute to unraveling the mechanisms underlying the disease severity and open new avenues for targeted therapeutic approaches.

## 2. Materials and Methods

### 2.1. Collection and Clinical Stratification of the Patients

From March to September 2020, 500 patients affected by COVID-19 were enrolled from various public hospitals located in Campania, a region of South Italy.

In Campania, at the beginning of March 2020, the most common Pango lineage was the B.1 (74%, sample count: 66); whereas, at the end of September 2020, B.1.177 was the most common strain (38%, sample count: 26) (https://covid19dashboard.regeneron.com, accessed on 1 June 2021). Of note, none of these two strains are classified as Variant-of-Interest or Variant-of-Concern by the Centers for Disease Control and Prevention (https://www.cdc.gov/coronavirus/2019-ncov/variants/variant-info.html, accessed on 1 June 2021). None of the enrolled patients were vaccinated at the time of the study.

Relevant features of the patients were summarized in Table 1. Among them, 121 (24.2%) hospitalized patients were subdivided into two age- and gender-matched subgroups: (i) 39 were severe cases that required mechanical ventilation (MV), and (ii) 82 were non-severe hospitalized patients that did not need MV. No differences in coexisting diseases between the two subgroups were observed. As a control group, we selected 379 individuals over 45 years of age that were positive for anti-SARS-CoV-2-IgG antibodies but did not show any symptoms. The antibody positivity was assessed by the Anti-SARS-CoV-2 Elecsys E2G 300 assay (Roche Diagnostics, Monza, Italy). A group of 283 healthy individuals (male 89.5%; mean age 45 ± 6.6 years) was also included in the study.

### 2.2. DNA Extraction, Quantification, and Library Preparation for Sequencing 

Genomic DNA of COVID-19 patients was extracted from peripheral blood using a Maxwell^®^ RSC Blood DNA Kit (Promega, Madison, WI, USA), and DNA concentration and purity were evaluated using a NanoDrop™ 8000 spectrophotometer. For library preparation, we measured the DNA concentration with a Qubit^®^ DNA Assay Kit with a Qubit^®^ 2.0 fluorometer (Life Technologies by Thermo Fisher Scientific, Waltham, MA, USA), and 1.0 μg of DNA was used for library preparation. Genomic DNA was sonicated to generate 180–250-bp fragments and then end-polished, A-tailed, and ligated with the full-length adapter with further PCR amplification. PCR products were purified using the AMPure XP system (Beckman Coulter, Brea, CA, USA) and quantified using the Agilent high-sensitivity DNA assay from the Agilent TapeStation 4200 system (Agilent Technologies, Santa Clara, CA, USA). The whole exome was captured with Agilent SureSelect Human All Exome V6 (Agilent Technologies, Santa Clara, CA, USA), and the sequencing was performed on an Illumina NovaSeq 6000 System (Illumina, San Diego, CA, USA).

### 2.3. Bioinformatic Analysis of Sequencing Data

The paired-end (2 × 150 bp) sequencing returned an average of 42 million raw reads per sample. After removing the sequencing artifacts, we retained, on average, 98.35% of the raw reads. The percentages of the bases with quality scores above 20 and above 30 (Q20 and Q30) were 97.7% and 93.8%, respectively. The cleaned reads were aligned versus the reference genome (GRCh37/hg19; RefSeq assembly accession: GCF_000001405.13), and mapping BAM files were obtained with BWA-mem (version 0.7.17) and SAMTools (version 1.8). On average, 99.89% of reads were mapped, and 20.47% of duplicate reads were removed with Picard (version 2.18.9). We covered 99.63% of the target regions, and the average sequencing depth of the target was 149.06x. Ninety-five point eight percent of the target regions were covered with at least 20 reads, which were sufficient for reliable variant calls. The SNVs and small insertions and deletions (INDELs) were detected with GATK HaplotypeCaller, and the functional annotation of the variants was performed with ANNOVAR. We obtained 189,625 raw SNVs and 32,089 raw INDELs per sample. We excluded off-target variants (e.g., intergenic, intronic, etc.) and single-nucleotide polymorphisms (SNPs) with allele frequencies greater than 1% in non-Finnish European populations of the 1000 Genomes Project, ExAC (v3), and GnomAD (v2.1.1) databases. To remove possible false positives, we also discarded variants falling in the genomic duplicated regions. The set of exonic variants was then filtered to remove synonymous SNVs.

### 2.4. Selection of CVID-Related Genes

To analyze the host genetic factors associated with COVID-19 severity, we performed a targeted WES analysis on 10 genes associated with CVID: *CD19*, *CD81*, *CR2*, *ICOS*, *MS4A1*, *NFKB1*, *NFKB2*, *PRKCD*, *TNFRSF13B*, and *TNFRSF13C* (Table 2). The selection of CVID-related genes was obtained by using the disease term “Common Variable Immunodeficiency” (Orphanet code, ORPHA: 1572) in the Human Phenotype Ontology database (https://hpo.jax.org/app/, accessed on 1 June 2021).

### 2.5. SNP Genotyping

The asymptomatic patients and the healthy controls were genotyped by the TaqMan^®^ SNP Genotyping Assay (C__89561246_10) for the single-nucleotide variant (SNV) rs61756766 (Applied Biosystems by Thermo Fisher Scientific, Waltham, MA, USA).

Allele and genotype frequencies were compared using the chi-square, Fisher’s exact, or Armitage tests. A two-sided *p* < 0.05 was considered statistically significant.

## 3. Results

### 3.1. Mutational Enrichment Analysis

The overall features of the 500 COVID-19 patients enrolled in the study are summarized in Table 1.

To identify the SNVs significantly enriched in severe COVID-19 patients compared to non-severe ones, we performed a targeted analysis of the WES data using an in silico panel composed of 10 genes associated with CVID, as derived from the Human Phenotype Ontology database (Table 2).

In particular, we compared the number of unrelated subjects carrying rare functional variants (alternative allele frequency ≤ 0.01; CADD > 20) in these 10 CVID-associated genes in each subgroup of 121 hospitalized patients, i.e., (i) 39 severe cases and (ii) 82 non-severe patients. In both subgroups, we identified SNVs in four genes (i.e., *TNFRSF13B*, *TNFRSF13C*, *CR2*, and *CD19*) (Table 3). Notably, we observed a statistically significant enrichment of SNVs in CVID genes among severe patients (8/38, 21.0%) compared to non-severe ones (6/82, 7.3%) (*p* = 0.03).

### 3.2. Identification of a Recurrent TNFRSF13C Variant Associated with COVID-19 Severe Phenotype

Among the CVID-associated genes, the *TNFRSF13C* gene showed the highest variant enrichment in severe patients (5/8, 62.5%). Of note, all these five patients carry a recurrent heterozygous rare variant, i.e., the SNV c.475C > T that leads to the amino acid change of H159Y (rs61756766, alternative allele frequency A = 0.7%, gnomAD European non-Finnish). In particular, we observed a statistically significant association between the rs61756766 SNV frequency and severity of COVID-19. Indeed, severe patients exhibited an increased frequency of the GA genotype (5/38, 13.2%) when compared to non-severe cases (1/82, 1.2%), with an odds ratio (OR) = 12.3 (*p* = 0.005). Comparable results were obtained by comparing the frequency of the heterozygous genotype between severe patients (5/38, 13.2%) and the asymptomatic ones (14/375, 3.7%) (OR = 3.9, *p* = 0.008). Accordingly, the frequency of the A allele was higher in severe cases (5/76, 6.6%) compared either to non-severe patients (1/164, 0.6%) (OR = 11.5, 95% CI: 1.3–100, *p* = 0.01) or to asymptomatic ones (14/750, 1.9%) (OR = 3.7, 95% CI: 1.3–10.6, *p* = 0.02) (Table 4).

## 4. Discussion

As for other viral infections, we can broadly classify the immune response to SARS-CoV-2 into two phases. At the early stage, a specific adaptive immune response is required to eradicate the virus and to prevent disease progression to severe stages. At the severe stage, the cytokine release syndrome and hyperactivation of the immune system are the main causes of life-threatening respiratory disorders leading to severe disease and, in some cases, to death [13].

Genetic differences are well-known to contribute to individual variations in the immune response to pathogens [2]. A few reports about the role of humoral immunity in COVID-19 have been published so far. A recent study in patients with primary antibody deficiencies and concurrent COVID-19 showed that patients with CVID (i.e., the presence of dysfunctional B lymphocytes) presented with a severe form of the disease requiring MV. Conversely, patients with agammaglobulinemia (i.e., the absence of B lymphocytes) presented mild symptoms. Thus, the different clinical course of COVID-19 in these two types of patients suggests a possible role of B lymphocytes in SARS-CoV-2-induced inflammation [11].

To understand the role played by the host genetic determinants involved in humoral immunity, we performed a WES analysis focused on CVID-associated genes by comparing SNV enrichment between severe and non-severe patients. Interestingly, we identified a rare variant in *TNFRSF13C* that was recurrent among severely affected patients. The *TNFRSF13C* gene encodes the BAFFR that is activated by BAFF, a TNF family member that supports the survival of B cells [12]. The c.475C > T transition results in the H159Y amino acid change within the highly conserved cytoplasmic domain of BAFFR [14]. This variant has already been associated with immune disorders, such as CVID, autoimmune lymphoproliferative syndrome-like disease, multiple sclerosis, Good’s syndrome, and Sjogren’s syndrome [15,16,17,18,19,20]. Additionally, it has been also associated with an increased risk of chronic lymphocytic leukemia and lymphoma [14,21].

Despite that the variant was not found to be associated with a change in BAFFR mRNA or protein expression [22], it has been proven to be a gain-of-function variant. Indeed, it increases ligand-independent BAFFR signaling, leading to enhanced NF-kB1 and NF-kB2 activation [14]. It is well-known in the role of NF-kB signaling in the induction of proinflammatory genes, including those encoding cytokines and chemokines [23]. Of note, it has been recently demonstrated that SARS-CoV-2 infection also induces an inflammatory phenotype in alveolar type 2 cells mediated by the hyperactivation of NF-kB signaling [24]. Thus, it is conceivable to hypothesize that the severe clinical course observed in COVID-19 patients carrying the rs61756766 AG genotype could be due to enhanced NF-kB signaling activation.

One limitation of this study might be the relatively low number of cases. However, it should be considered that our cohort of cases was derived from a careful selection to define homogeneous groups of patients. Furthermore, all the studied individuals were of Italian origin and residents in the same geographical area, which limited the problem relative to the population structure. Replication studies are needed to further validate our findings. Additionally, we herein focused on the H159Y variant, since it is the most frequently detected rare SNV in severe patients and since it happens to have strong biological support from the literature. Nevertheless, other non-tested variants could be associated with severe COVID-19. Thus, we encourage other research groups to carry out further genetic studies considering our preliminary results. 

Our findings are relevant not only from the perspective of the pathogenesis of this condition but, also, for therapeutic implications. Indeed, the role of B cells in determining inflammatory disorders is also supported by the evidence of the beneficial effects of B-cell depletion in the management of these diseases [11,20]. Thus, the identification of the humoral immunity risk factors associated with a worse COVID-19 outcome could provide a rationale for individualized management based on B-cell therapy.

## Figures and Tables

**Table 1 genes-12-00881-t001:** General features of the affected patients.

	Severe Cases		Non-Severe Cases			Asymptomatic Cases		
	*n* = 39	%	*n* = 82	%	* *p*	*n* = 379	%	^ *p*
Age								
years, mean (standard deviation)	62.5 (13.3)		62.2 (15.5)		0.90	60.9 (10.7)		0.4
Gender—no. (%)								
Male	22	56.4	52	63.4		180	47.4	
Female	17	43.6	30	36.6	0.46	199	52.5	0.29
Previous coexisting disease—no. (%)								
0–2	25	64.1	48	58.5		na		
≥3	7	17.9	19	23.2	0.47	na		
Unknown	7	17.9	15	18.3				

* Severe cases vs. Non-severe cases, chi-square test. ^ Severe cases vs. Asymptomatic cases, chi-square test. na: not available.

**Table 2 genes-12-00881-t002:** Genes associated with the disease term “Common Variable Immunodeficiency”.

Gene Symbol	Approved Name HGNC	Location	Phenotype	Phenotype MIM Number	Inheritance
*CD19*	CD19 molecule	16p11.2	Immunodeficiency, common variable, 3	613493	AR
*CD81*	CD81 molecule	11p15.5	Immunodeficiency, common variable, 6	613496	AR
*CR2*	Complement C3d receptor 2	1q32.2	Immunodeficiency, common variable, 7	614699	AR
*ICOS*	Inducible T cell costimulator	2q33.2	Immunodeficiency, common variable, 1	607594	AR
*MS4A1*	Membrane spanning 4-domains A1	11q12.2	?Immunodeficiency, common variable, 5	613495	AR
*NFKB1*	Nuclear factor-kappa B subunit 1	4q24	Immunodeficiency, common variable, 12	616576	AD
*NFKB2*	Nuclear factor-kappa B subunit 2	10q24.32	Immunodeficiency, common variable, 10	615577	AD
*PRKCD*	Protein kinase C delta	3p21.1	Autoimmune lymphoproliferative syndrome, type III	615559	AR
*TNFRSF13B*	TNF receptor superfamily member 13B	17p11.2	Immunodeficiency, common variable, 2	240500	AD, AR
*TNFRSF13C*	TNF receptor superfamily member 13C	22q13.2	Immunodeficiency, common variable, 4	613494	AR

HGNC, HUGO Gene Nomenclature Committee. MIM number, entry number of the phenotypes cataloged in Online Mendelian Inheritance in Man database (OMIM, https://www.ncbi.nlm.nih.gov/omim/, accessed on 1 June 2021). AR: autosomal recessive; AD: autosomal dominant.

**Table 3 genes-12-00881-t003:** Genetic variants identified in CVID-related genes.

Subgroup	Patient ID	Gene Symbol	HGVS cDNA-Level Nomenclature	HGVS Protein-Level	RefSeq ID	AF Gnomad ^§^	Genotype	CADD Score	InterVar
Severe patients	R_41	*TNFRSF13C*	c.475C > T	p.H159Y	rs61756766	0.0075	0/1	27.6	VUS
	R_42	*TNFRSF13B*	c.542C > A	p.A181E	rs72553883	0.0064	0/1	22.8	Benign
	R_48	*TNFRSF13C*	c.475C > T	p.H159Y	rs61756766	0.0075	0/1	27.6	VUS
	R_69	*TNFRSF13C*	c.475C > T	p.H159Y	rs61756766	0.0075	0/1	27.6	VUS
	R_158	*TNFRSF13C*	c.475C > T	p.H159Y	rs61756766	0.0075	0/1	27.6	VUS
	R_167	*TNFRSF13C*	c.475C > T	p.H159Y	rs61756766	0.0075	0/1	27.6	VUS
	R_168	*CR2*	c.1676G > A	p.G559E	rs143614333	0.0008	0/1	24.8	VUS
	R_176	*CR2*	c.1021C > T	p.R341C	rs529311780	0.0001	0/1	28.5	VUS
Non-severe patients	R_15	*TNFRSF13B*	c.310T > C	p.C104R	rs34557412	0.0055	0/1	25.9	Benign
	R_46	*CR2*	c.1676G > A	p.G559E	rs143614333	0.0008	0/1	24.8	VUS
	R_99	*CR2*	c.1723C > T	p.R575C	rs758797015	0.0000	0/1	32.0	VUS
	R_127	*TNFRSF13B*	c.492C > G	p.Y164 *	rs72553882	0.0001	0/1	37.0	Pathogenic
	R_140	*CD19*	c.5C > T	p.P2L	rs766808956	0.0000	0/1	26.9	VUS
	R_143	*TNFRSF13C*	c.475C > T	p.H159Y	rs61756766	0.0075	0/1	27.6	VUS

*, the curent nomencalture for the stopgain variants. ^§^ AF, alternative allele frequency in European non-Finnish. Statistically significant enrichment of mutations in CVID genes among severe patients (8/38, 21.0%) compared to non-severe ones (6/82, 7.3%) (chi-square test, *p* = 0.03). VUS, variant of uncertain significance. NCBI RefSeq transcript for each gene: *CD19*, NM_001178098; *CR2*, NM_001006658; *TNFRSF13B*, NM_012452; *TNFRSF13C*, NM_052945.

**Table 4 genes-12-00881-t004:** Genetic association of rs61756766 SNV in *TNFRSF13C* with severe cases.

	Severe Cases	Non-Severe Cases	Asymptomatic Cases	Population
	*N* = 38	*N* = 82	*N* = 375	*N* = 283
Genotype—no. (%)				
GG	33 (86.8)	81 (98.8)	361 (96.3)	269 (95.0)
GA	5 (13.2)	1 (1.2)	14 (3.7)	14 (5.0)
AA	0 (0.0)	0 (0.0)	0 (0.0)	0
^ Ptrend		0.005		
^ Common OR		12.3		
^§^ Ptrend			0.008	
^§^ Common OR			3.9	
^#^ Ptrend				0.04
^#^ Common OR				2.9
Allele—no. (%)				
G	71 (93.4)	163 (99.4)	736 (98.1)	552 (97.5)
A	5 (6.6)	1 (0.6)	14 (1.9)	14 (2.5)
* Pallele		0.01		
* OR (95% CI)		11.5 (1.3–100)		
° Pallele			0.02	
° OR (95% CI)			3.7 (1.3–10.6)	
^$^ Pallele				0.06
^$^ OR (95% CI)				2.8 (0.97–7.9)

^ Armitage trend test: Severe cases vs. Non-severe cases; ^§^ Armitage trend test: Severe cases vs. Asymptomatic cases; ^#^ Armitage trend test: Severe cases vs. Population; * Fisher’s exact test: Severe cases vs. Non-severe cases; ° Fisher’s exact test: Severe cases vs. Asymptomatic cases; ^$^ Fisher’s exact test: Severe cases vs. Populations.

## Data Availability

The data that supported the findings of this study are available from the corresponding author upon reasonable request.

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
