# Peer review of "The TNFRSF13C H159Y Variant Is Associated with Severe COVID-19: A Retrospective Study of 500 Patients from Southern Italy"

_genes, 2021, doi:10.3390/genes12060881_

Round 1

Reviewer 1 Report

Russo and colleagues described an association of TNFRSF13C H159Y variant with severe COVID-19 using well-defined cohorts of patients and controls from Southern Italy. As already acknowledged by the authors, the main limitation of this study is the relative small size of the severe patient subgroup (n = 38). Nonetheless, (i) the use of other patient subgroups and controls from the same geographical region and (ii) strong biological support from the literature made the finding potentially significant. This report is likely to stimulate replication studies in other populations.

I have some minor comments:

  1. Gene symbols should be italicized throughout.
  2. In Materials and Methods: it would be helpful to include information about how the 10 genes associated with CVID were selected; and better to change Table S1 to a main table.
  3. Hys159Tyr should be p.His159Tyr.
  4. Lines 1224-225: “Additionally, we herein focused on the H159Y variant since it has been functionally characterized in several independent studies”. It seems to me that you did so primarily because H159Y was the most frequently detected rare SNV in the severe patients; and it happens to have a strong biological support from the literature.

Author Response

We thank the reviewer for his/her revision. We amended the text as requested:

- the gene symbols have been italicized throughout the text.

- we improved the “Materials and Methods” section by providing information about the selection of the 10 CVID-related genes. Additionally, we moved Table S1 to the main text. In the revised version, Table S1 has been replaced by Table 2.

- we corrected the acronym of the amino acid Histidine (p.His159Tyr)

- we amended the text at lines 1224-225 as follows: ” Additionally, we herein focused on the H159Y variant since it is the most frequently detected rare SNV in severe patients, and since it happens to have strong biological support from the literature”.

Reviewer 2 Report

Review article: The TNFRSF13C H159Y variant is associated with severe COVID-19: a retrospective study of 500 patients from southern Italy.

Comments:     In this study the authors describe the nucleotide identity of 10 genes that are involved in humoral immunity and associated with the risk of developing severe COVID-19 symptoms, including the need for mechanical ventilation. Specifically, the authors found a recurrent TNFRSF13C variant associated with COVID-19 phenotype. In wonder whether the authors know what viral strains affect these individuals? Also, any of these individuals were vaccinated?

I wonder whether the authors could apply this test to predict immune response against COVID-19? For example, run this test in healthy individuals and then, in case of disease, follow up the patient as they have the variant, then more likely to develop severe COVID-19?

If the authors don’t know the circulating SARS-cov2 strains in the geographic region they describe, then they could find that information in the https://www.gisaid.org/ . Perhaps that complementary information could enhance the overall quality of this work.

Specific comments:

Line 113. Reference genome: provide accession number

Author Response

We thank the reviewer for his/her revision. We modified the manuscript according to the suggestions provided by the present reviewer.

Q1. Comments: In this study the authors describe the nucleotide identity of 10 genes that are involved in humoral immunity and associated with the risk of developing severe COVID-19 symptoms, including the need for mechanical ventilation. Specifically, the authors found a recurrent TNFRSF13C variant associated with COVID-19 phenotype. In wonder whether the authors know what viral strains affect these individuals? Also, any of these individuals were vaccinated?

A1. We added the following period in the Material and Methods section:

“During that period [From March to September 2020], in Campania, the most common Pango lineage was the B.1 (74%, sample count: 66) at beginning of March 2020, whereas, at the end of September 2020, the most common strain was the B.1.177 one (38%, sample count: 26) (https://covid19dashboard.regeneron.com). Of note, none of these two strains were classified as Variant-of-Interest or Variant-of-Concern by the Centers for Disease Control and Prevention (https://www.cdc.gov/coronavirus/2019-ncov/variants/variant-info.html). None of the enrolled patients were vaccinated at the time of the study”.

Q2. I wonder whether the authors could apply this test to predict immune response against COVID-19? For example, run this test in healthy individuals and then, in case of disease, follow up the patient as they have the variant, then more likely to develop severe COVID-19?

A2. We thank the reviewer for this comment. In this study, we focused on a specific subset of genes involved in Mendelian diseases affecting humoral immunity. Despite it is conceivable to hypothesize that genetic variants in these genes might have a highly penetrant effect on the risk of developing severe COVID-19, we have to consider that COVID-19 is a multifactorial disorder. Thus, the contribution of genetic host factors cannot be limited to the identification of single genetic variants. Instead, it seems more appropriate to generate a polygenic risk score, including many common and rare risk variants, to be used as genetic testing to predict the adverse outcome of COVID-19.

Q3. If the authors don’t know the circulating SARS-cov2 strains in the geographic region they describe, then they could find that information in the https://www.gisaid.org/. Perhaps that complementary information could enhance the overall quality of this work.

A3. See answer to question 1 (A1).

Q4. Line 113. Reference genome: provide accession number

A4. We provide the accession number of the reference genome (GRCh37/hg19; RefSeq assembly accession: GCF_000001405.13) as requested.